# Immunogenicity and Safety of an Inactivated Quadrivalent Influenza Vaccine Administered Concomitantly with a 23-Valent Pneumococcal Polysaccharide Vaccine in Adults Aged 60 Years and Older

**DOI:** 10.3390/vaccines12080935

**Published:** 2024-08-22

**Authors:** Zhongkui Zhu, Jianwen Sun, Yan Xie, Xi Lu, Wanqin Tang, Yanwei Zhao, Lu Shen, Huaxian Liu, Yang Yu, Siliang Zhou, Liqun Huo, Peng Jiao, Xiaoli Jiang

**Affiliations:** 1Department of Immunization Program, Taizhou City Center for Disease Control and Prevention, Taizhou 225300, China; duntianzhu@163.com (Z.Z.); xytzcdc@163.com (Y.X.); twqpinfan@163.com (W.T.); hlxysl@163.com (L.S.); 18651144798@163.com (H.L.); 15996066068@163.com (Y.Y.); 2Department of Medical Affairs, Sinovac Biotech Co., Ltd., Beijing 100089, China; 17332227279@163.com (J.S.); lux5607@sinovac.com (X.L.); zhaoyw@sinovac.com (Y.Z.); zhousl7810@sinovac.com (S.Z.); huolq@sinovac.com (L.H.); jiaop@sinovac.com (P.J.)

**Keywords:** inactivated quadrivalent influenza vaccine, 23-valent pneumococcal polysaccharide vaccine, concomitant administration, immunogenicity, safety, older adults

## Abstract

The inactivated quadrivalent influenza vaccine (IIV4) and the 23-valent pneumococcal polysaccharide vaccine (PPSV23) have been administered for years and could be administered concomitantly if necessary. However, the immunogenicity and safety of the concomitant administration of these two vaccines have not been well documented, especially in the Chinese population. In this study, 480 participants aged 60 years and older were randomly assigned to the concomitant administration group (C group) or the separate administration group (S group) to receive IIV4 and PPSV23 either concomitantly or separately. Blood samples were collected before and 28 days after each vaccination. The antibodies against four influenza virus strains and twenty-three pneumococcus serotypes were tested. The results showed that the geometric mean titer (GMT) ratios (C group to S group) for the four influenza strains ranged from 0.72 to 0.95, with the lower limits of the 95% confidence intervals (CIs) ranging from 0.51 to 0.75, and the geometric mean concentration (GMC) ratios for the 23 pneumococcal serotypes ranged from 0.80 to 1.00, with the lower limits of 95% CIs ranging from 0.67 to 0.86. All values met the predefined criteria for non-inferiority. The incidence of adverse events was 0.63% in the C group and 1.56% in the S group. No serious adverse events were observed. In conclusion, the immunogenicity of the concomitant administration of IIV4 and PPSV23 was non-inferior to that of the separate administration, and the safety profile was favorable in adults aged 60 years and older in China.

## 1. Introduction

Influenza virus and *Streptococcus pneumoniae* (SP) are two common respiratory pathogens that can lead to acute respiratory infections and severe illness [1,2,3]. Influenza virus can cause Influenza and lead to severe complications, including pneumonia, myocarditis, encephalitis, myositis and rhabdomyolysis [4]. A systematic review and meta-analysis found that the pooled attack rate of influenza was 22.5% for unvaccinated children and 10.7% for unvaccinated adults [5]. It is estimated that there are about 1 billion cases of influenza worldwide annually, of which 3–5 million are severe, and 290,000–650,000 influenza-related respiratory deaths [6]. *Streptococcus pneumoniae* could colonize the nasopharyngeal mucosa, and spread to non-sterile sites, causing non-invasive pneumococcal disease (NIPD) such as non-bacteraemic pneumonia, acute otitis media, and sinusitis. It can also spread to sterile sites, leading to invasive pneumococcal disease (IPD) like bacteraemic pneumonia and meningitis [7]. It was estimated that in 2017, there were 1,166,374 deaths globally due to lower respiratory tract infections caused by pneumococcus pneumoniae. Of these, 380,931 deaths occurred in children under 5 years of age and 456,096 in older adults over 70 years of age [8]. The Active Bacterial Core surveillance (ABCs) system in the U.S. reported that nearly 70% of IPD cases in 2021 occurred in adults aged 50 years and older [9].

Influenza vaccines, including trivalent influenza vaccine (TIV) and quadrivalent influenza vaccine (QIV), are approved for the prevention of influenza [10,11,12] and the World Health Organization (WHO) recommends annual influenza vaccination to individuals with particular influenza risk, such as health workers, older adults, individuals with comorbidities and pregnant individuals [4]. A meta-analysis of test-negative design case–control studies reported that the matched vaccine effectiveness was 44.38% and the mismatched vaccine effectiveness was 20.00% [13]. PPSV23 is recommended for older adults in countries such as Canada and the UK [14,15]. A meta-analysis of randomized controlled trials (RCTs) reported the protective efficacy of PPSV23 against definitive pneumococcal pneumonia was 74% [16]. In China, the influenza vaccine is recommended to priority populations, such as health workers, adults aged 60 years and older, individuals with underlying conditions, pregnant women, and children under 6 years of age [17,18], and PPSV23 is recommended for specific at-risk populations such as adults aged 60 years and older [18,19].

Some studies have found that compared to vaccination with only the influenza vaccine or PPSV23, vaccination with both vaccines would provide better effectiveness [20,21,22]. And previous studies reported that the concomitant administration of TIV and PPSV23 did not affect their immunogenicity or safety [23,24]. However, clinical evidence on the concomitant administration of quadrivalent influenza vaccine (QIV) and PPSV23 is lacking. There was also no previous evidence in the Chinese population. Therefore, a clinical study was conducted on the immunogenicity and safety of concomitant vaccination with the inactivated quadrivalent influenza vaccine (IIV4) and the 23-valent pneumococcal polysaccharide vaccine. The results are reported as follows.

## 2. Materials and Methods

### 2.1. Study Design and Participants

This was a phase IV, open-label, randomized, positive controlled non-inferiority trial (NCT05471531) to evaluate the immunogenicity and safety of an inactivated quadrivalent influenza vaccine (IIV4) administered concomitantly with a 23-valent pneumococcal polysaccharide vaccine (PPSV23) during the 2021–2022 influenza season. It was conducted at the single site of Taizhou City Hospital of Traditional Chinese and Western Medicine.

Participants aged 60 years and older who could understand and voluntarily sign the informed consent form were eligible for inclusion. Recruitment notices were issued to participants who might meet the inclusion criteria. Key exclusion criteria included participants who (1) had already received a 2021–2022 seasonal influenza vaccine before screening; (2) had received PPSV23 within the last five years; (3) had a history of severe allergic reactions to vaccines; (4) had uncontrolled epilepsy or other serious neurological disorders; (5) were experiencing a fever, an acute exacerbation of chronic diseases, uncontrolled severe chronic diseases or an acute illness; (6) had any other risk factors deemed unsuitable for participation by the investigator. Full inclusion and exclusion criteria are listed in the Appendix A.

All necessary documents for this trial were reviewed and approved by the Ethics Committee of the Taizhou City Hospital of Traditional Chinese and Western Medicine (2021-Ethics Review-11). The clinical trial was conducted in accordance with the principles of the Declaration of Helsinki and the International Council for Harmonisation Good Clinical Practice. All participants provided their written informed consent before screening.

### 2.2. Investigational Vaccines

The influenza vaccine (Split Virion), inactivated, quadrivalent and the 23-valent pneumococcal polysaccharide vaccine used in this study were developed and manufactured by Sinovac Biotech Co., Ltd. (Beijing, China) Each dose of IIV4 contained 15 µg of hemagglutinin from each of the four influenza strains recommended by the WHO for the 2021–2022 season (A/Victoria/2570/2019 (H1N1) pdm09-like virus; A/Cambodia/e0826360/2020 (H3N2)-like virus; B/Washington/02/2019 (B/Victoria lineage)-like virus; B/Phuket/2027/2013 (B/Yamagata lineage)-like virus). Each dose of PPSV23 contained 25 µg of capsular polysaccharides for each of the 23 serotypes (i.e., 1, 2, 3, 4, 5, 6B, 7F, 8, 9N, 9V, 10A, 11A, 12F, 14, 15B, 17F, 18C, 19A, 19F, 20, 22F, 23F and 33F).

### 2.3. Procedures

A total of 480 participants were enrolled and randomly assigned to the concomitant administration group (C group) or the separate administration group (S group); 160 participants in the C group received one dose of IIV4 and one dose of PPSV23 concomitantly on day 0, 160 participants in the separate administration subgroup 1 (S1 subgroup) received a single dose of IIV4 on day 0 and a single dose of PPSV23 on day 28, 160 participants in the separate subgroup 2 (S2 subgroup) received one dose of PPSV23 on day 0 and one dose of IIV4 on day 28. All participants received PPSV23 in the left arm and IIV4 in the right arm. Blood samples were drawn before and at 28 days after each vaccination: on day 0 and day 28 for the C group; on days 0, 28 and 56 for the S group. Antibodies for four influenza virus strains, and twenty-three pneumococcus serotypes were tested before and at 28 days after the corresponding vaccine administration. For safety assessment, participants were required to report any adverse events within 28 days following each vaccination. Serious adverse events were recorded throughout the trial. Adverse events were graded according to the guidelines released by the China National Medical Products Administration (NMPA) [25].

### 2.4. Outcomes and Endpoints

The primary endpoint of the study was the geometric mean titers (GMT) of hemagglutination inhibition (HI) antibodies against four influenza virus strains at 28 days after the concomitant administration of IIV4. The secondary endpoints were the seroprotection rates (SPRs), seroconversion rates (SCRs) and geometric mean fold rises (GMFRs) of antibodies against four influenza virus strains after the concomitant administration of IIV4; the GMTs, SCRs, GMFRs and SPRs of antibodies against four influenza virus strains at 28 days after the separate administration of IIV4; and the geometric mean concentrations (GMCs), the 2-fold increase rates and GMFRs of pneumococcal antibodies against 23 serotypes at 28 days after the concomitant and separate administration of PPSV23. The safety endpoints were the incidences of adverse events and adverse reactions within 7 days and 28 days after the concomitant or separate administration of IIV4 and PPSV23. The SPR of influenza HI antibodies was defined as the proportion of participants with an HI antibody titer of 1:40 or higher. The SCR of IIV4 was defined as the percentage of participants with either a pre-vaccination HI titer < 1:10 and a post vaccination HI titer > 1:40, or a pre-vaccination HI titer > 1:10 and at least a four-fold increase after vaccination. The 2-fold increase rate of PPSV23 was defined as the percentage of participants with a ≥2-fold increase in post-vaccination antibody concentration compared to the pre-vaccination concentration.

### 2.5. Statistical Analysis

#### 2.5.1. Sample Size

The primary objective was to assess whether the immunogenicity of IIV4 in the co-administration group (C group) was non-inferior to that in the separate administration group (S group) in terms of antibody GMTs for four influenza virus strains at 28 days after vaccination. And the secondary objective was to explore whether the immunogenicity of PPSV23 in the C group was non-inferior to that in the S group in terms of antibody GMCs for 23 pneumococcal serotypes 28 days after vaccination. For the non-inferiority standard, the lower limits of the 95% confidence intervals (CIs) for the GMT ratios (GMT_c-group_/GMT_s-group)_ for all four strains should be greater than 0.5. In other words, the lower limits of the 95% CIs for the differences in GMTs after logarithmic transformation should be greater than −0.301. Based on the results in the phase III clinical trial for IIV4, the standard deviation after logarithmic transformation in adults aged 60 and older was 0.7. Assuming N_c-group_: N_s-group_ = 1:2 (N_c-group_: N_s1-group_: N_s2-group_ = 1:1:1), a one-sided α = 0.025 and a total power of 80%, the required effective sample size in each arm was calculated to be 142 participants using the NCSS-PASS software (version: 15.0.5). Accounting for a 10% dropout rate, there would be 160 people in each arm, resulting in a total of 480 participants.

#### 2.5.2. Immunogenicity Assessment

For the antibody GMTs, GMCs and GMFRs for each group, the geometric mean and 95% CIs were calculated, and the differences between groups were tested using log-transformed analysis of variance. For the SCRs and SPRs of IIV4 and the 2-fold increase rate of PPSV23, the two-sided 95% CIs were calculated, and differences between groups were tested using the chi-squared test/Fisher’s exact test. Non-inferiority comparisons were analyzed using the analysis of covariance (ANCOVA) model. Logarithmically transformed post-vaccination antibody GMTs and GMCs were defined as the dependent variables and the logarithmically transformed pre-vaccination antibody GMTs or GMCs were the covariates. The least square mean and 95% CI for the GMT or GMC ratio between groups were calculated. Non-inferiority was demonstrated if the lower limit of 95% CI for the GMT or GMC ratio (C group/S group) was greater than 0.5. The immunogenicity endpoints were assessed in the per-protocol population, which included participants who completed all visits, received the full-schedule vaccination and had the valid serum results without any protocol deviation.

#### 2.5.3. Safety Assessment

The incidences and severities of adverse events and adverse reactions were calculated in the safety population, which included all participants who had received at least one dose of the vaccines. Participants who received the wrong vaccine were analyzed on the basis of the vaccines they actually received, according to the All Participants as Treated (ASaT) principle. Differences between groups were tested using Fisher’s exact test.

### 2.6. Serological Assays

The antibody titers against four influenza vaccine-associated strains (A/Victoria/2570/2019 (H1N1) pdm09-like virus; A/Cambodia/e0826360/2020 (H3N2)-like virus; B/Washington/02/2019 (B/Victoria lineage)-like virus; B/Phuket/2027/2013 (B/Yamagata lineage)-like virus)) were measured using a hemagglutination inhibition (HI) assay following the WHO Manual for the Laboratory Diagnosis and Virological Surveillance of Influenza [26]. The international reference standards of influenza antigens were obtained from the National Institute for Biological Standards and Control (NIBSC, London, UK). Pneumococcal antibodies were quantified by enzyme linked immunosorbent assay (ELISA) according to the WHO manual for the quantitation of *Streptococcus pneumoniae* serotype-specific IgG (Pn PS ELISA) [27]. The pneumococcal polysaccharide standards were sourced from the American Type Culture Collection (ATCC, Manassas, VA, USA).

### 2.7. Use of Artificial Intelligence-Assisted Technology

After completing the draft of the manuscript, we used OpenAI ChatGPT 4.0 tool to find and correct grammatical errors in the manuscript.

## 3. Results

### 3.1. Study Population

From November 2021 to May 2022, a total of 480 participants were enrolled and randomly assigned to receive IIV4 and PPSV23 concomitantly (C group), IIV4 alone (S1 subgroup) or PPSV23 alone (S2 subgroup) on day 0. Additionally, 151 participants in the S1 subgroup received PPSV23 and 151 participants in the S2 subgroup received IIV4 on day 28 (Figure 1).

### 3.2. Demographic Characteristics

The mean age of the participants was 68.59 ± 5.33 years and the gender ratio was balanced. All participants were of Han ethnicity. The mean Body Mass Index (BMI) was 25.00 ± 3.28, primarily ranging from 20 to 30. No significant differences in demographic characteristics were observed either between the concomitant administration group (C group) and the separate administration group (S group) (Table 1) or between the S1 and S2 subgroups (Appendix A).

### 3.3. Immunogenicity

#### 3.3.1. The Immunogenicity of IIV4 and Non-Inferiority of Concomitant Administration

At baseline, the geometric mean titers (GMTs) of pre-vaccination hemagglutination inhibition (HI) antibody for four influenza virus strains ranged from 12.71 to 59.93 in the C group and from 12.15 to 71.66 in the S group. The seroprotection rates (SPRs) ranged from 12.82% to 82.05% in the C group and from 17.23% to 84.12% in the S group. The pre-vaccination antibody levels against the A/H1N1 and B/Victoria strains were lower. The GMTs against the A/H3N2 and B/Yamagata strains in the S group were higher than those in the C group (*p* = 0.0420 and *p* = 0.0292, respectively) (Table 2).

At 28 days after IIV4 vaccination, the HI antibody levels increased. The GMTs of post-vaccination HI antibodies for four influenza virus strains, adjusted by the analysis of covariance (ANCOVA) model, ranged from 91.51 to 567.82 in the C group and from 96.20 to 793.75 in the S group. The highest antibody GMT was observed against the A/H1N1 strain, and the lowest one was against the B/Victoria strain. The GMT ratios (GMT_c-group_/GMT_s-group_) for the four influenza strains ranged from 0.72 to 0.95, with the lower limits of the 95% CIs ranging from 0.51 to 0.75, which all met the non-inferiority criteria (Figure 2). Except for the geometric mean fold rise (GMFR) for the A/H1N1 strain, the post-vaccination SPRs, seroconversion rates (SCRs) and GMFRs were similar between the C and the S group. The GMFR for the A/H1N1 strain in the C group was lower than that in the S group (*p* = 0.0445) (Table 3). However, compared with the S1 subgroup, the influenza antibody GMFRs for all four strains in the C group were similar (Appendix A).

The pre-vaccination GMTs and SPRs for the four influenza virus strains were similar among the C, S1 and S2 subgroup, except for the GMT for yamagata lineage (Appendix A). The GMT for Yamagata lineage in the S2 subgroup was higher than those in the C group and S1 subgroup. Except for the GMFR against B/Yamagata strain, the post-vaccination SPRs, SCRs, adjusted GMT and GMFRs were similar among three groups. The GMFR for B/Yamagata strain in the S1 subgroup was higher than that in the S2 subgroup (*p* = 0.007), but comparable to that in the C group (Appendix A). The post-vaccination GMT ratios (GMT_S2-subgroup_/GMT_S1-subgroup_) for the four strains, adjusted by the ANCOVA model, ranged from 0.75 to 1.05, with the lower limits of the 95% CIs ranging from 0.58 to 0.72, which all met the equivalence criteria (Appendix A).

#### 3.3.2. The Immunogenicity of PPSV23 and Non-Inferiority of Concomitant Administration

The pre-vaccination antibody GMCs for 23 serotypes ranged from 0.45 to 4.92 µg/mL in the C group and from 0.44 to 5.22 µg/mL in the S group, as shown in Table 4. The lowest GMCs were observed for serotype 3 in both groups, while the highest GMCs were observed for serotype 19A in the C group and serotype 14 in the S group. Except for serotypes 12F and 15B, there were no significant differences in the GMCs between the C group and the S group. The GMC for serotype 12F in the C group was higher than that in the S group (*p* = 0.0002), while the GMC for serotype 15B in the C group was lower than that in the S group (*p* = 0.0374).

At 28 days after PPSV23 vaccination, the adjusted pneumococcal antibody GMCs ranged from 0.96 to 26.30 µg/mL in the C group and from 1.03 to 29.59 µg/mL in the S group. The lowest GMCs were still for serotype 3 in both groups and the highest GMCs were for serotype 15B in the C group and serotype 33F in the S group. GMT ratios (GMT_c-group_/GMT_s-group_) for the 23 pneumococcal serotype ranged from 0.80 to 1.00, with the lower limits of 95% CIs ranging from 0.67 to 0.86, which all met the non-inferiority criteria (Figure 3).

The two-fold increase rates of pneumococcal antibodies for 23 serotypes at 28 days after PPSV23 vaccination ranged from 43.59% to 93.59% in the C group and from 51.88% to 94.20% in the S group (Table 5). Compared with the S group, the two-fold increase rates for serotypes 19A and 22F were lower in the C group (*p* = 0.0024 and *p* = 0.0012, respectively), while the two-fold increase rates for the other 21 serotypes were similar. The pneumococcal antibody GMFRs for 23 serotypes after PPSV23 vaccination ranged from 2.15 to 7.83 in the C group and from 2.33 to 10.00 in the S group. For serotypes 6B, 12F, 19A, 22F and 33F, the pneumococcal antibody GMFRs in the C group were lower than those in the S group, while for the other 18 serotypes, the GMFRs were similar between the two groups. Furthermore, when compared with the S2 subgroup, the GMFRs for all 23 serotypes in the C group were similar (Appendix A).

Furthermore, the pre-vaccination GMCs for the 23 serotypes were similar among the C, S1, and S2 subgroups, except for serotype 12F, which was lower in the S1 subgroup than in the C and S2 groups (Appendix A). After vaccination, the antibody levels were almost similar among the three groups (Appendix A). Compared with the S2 subgroup, all the post-vaccination GMFRs in the C group were similar. However, the adjusted GMC for serotype 6B, the 2-fold increase rate for serotype 19A, and both the adjusted GMC and 2-fold increase rate for serotype 22F were lower in the C group. For the remaining serotypes, the post-vaccination adjusted GMCs and 2-fold increase rates were comparable between the two groups. Notably, the 2-fold increase rate and GMFR for serotype 12F in the S1 subgroup were higher than those in the S2 subgroup (*p* = 0.0152 and *p* = 0.0007, respectively). For the other serotypes, the adjusted GMCs, 2-fold increase rates and GMFRs were similar between the S1 subgroup and the S2 subgroup (Appendix A). The post-vaccination GMC ratios (GMC_S1-subgroup_/GMC_S2 subgroup_) for 23 serotypes, adjusted by the ANCOVA model, ranged from 0.78 to 1.00. The lower limits of 95% CIs ranged from 0.66 to 0.86, which all met the equivalence criteria (Appendix A).

### 3.4. Safety

The overall incidence of adverse events within 28 days after vaccination was 0.63% (1/160) in the C group and 1.56% (5/320) in the S group (Table 6). Adverse events were primarily Grade 1. No Grade 3 or higher adverse events or serious adverse events were observed in the study. The total incidence of local adverse events was 0.63% (1/160) in the C group and 1.25% (4/320) in the S group. The main local symptom was pain at the vaccination site. Systemic adverse events occurred in one participant from the S group, who experienced fever and fatigue at Grade 2 severity and a Grade 1 cough. The incidence and severity of any type of adverse event were comparable between the C group and the S group.

## 4. Discussion

The inactivated quadrivalent influenza vaccine (IIV4) and 23-valent pneumococcal polysaccharide vaccine (PPSV23) have been used in China for several years. However, there is a lack of clinical study data exploring the immunogenicity and safety of concomitant administration of IIV4 and PPSV23, especially among the elderly population aged 60 years and older. In response, in 2021, the Taizhou City Center for Disease Control and Prevention conducted an immunogenicity and safety study on the concomitant administration of IIV4 and PPSV23 in adults aged 60 years and older. The results demonstrated that the immunogenicity of concomitant administration of IIV4 and PPSV23 was non-inferior to that of their separate administration, as the GMTs for all four influenza strains and GMCs for all twenty-three pneumococcal serotypes at 28 days after concomitant administration were non-inferior to those after separate administration. The concomitant administration was also well tolerated, with low incidences of adverse events. The incidence and severity of each type of adverse event were comparable between the C group and the S group, indicating that the safety profiles of the two vaccination methods were similar.

In China, IIV4 is recommended for individuals aged 3 years and older, and PPSV23 is recommended for individuals aged 2 years and older who are at risk of pneumococcal infections [17,18,19]. Given the significant impact of influenza and pneumococcal diseases on public health, as well as their associated social and economic burden, some provinces and cities in China provide free vaccination with IIV4 and PPSV23 for the elderly, and recommend concomitant vaccination with these two vaccines. Concomitant administration could provide more comprehensive protection, reduce missed opportunities for vaccination (MOV) and improve coverage and timeliness of vaccination [28,29]. Our study is the first study to investigate the immunogenicity and safety of these two vaccines in the Chinse population. The findings provided crucial clinical evidence to support the concomitant vaccination policies for these two vaccines.

The results of two previous studies also showed that the concomitant administration of IIV4 and PPSV23 had favorable immunogenicity and safety profiles, which aligned with the conclusions of this study [30,31]. However, these two studies did not assess all 23 serotypes covered by the PPSV23; only a portion of the serotypes were tested. In contrast, our study tested all 23 serotypes and found that antibody levels for all 23 serotypes after concomitant vaccination were non-inferior to those after separate vaccination, the evidence of which was more scientific to support the concomitant vaccinations of these two vaccines.

Our study also demonstrated that IIV4 and PPSV23, whether administered alone or concomitantly, could exhibit favorable immunogenicity. The SCRs, GMIs and SPRs of influenza HI antibodies for IIV4 after concomitant or separate administration in this study all exceeded the corresponding evaluation criteria of the European Medicines Agency (EMA) [32] and the US Food and Drug Administration (FDA) [33], which was consistent with previous studies in different influenza seasons [34,35]. PPSV23 is known to cover approximately 85% of the pathogenic serotypes prevalent in China [19]. And In China, pneumonia diseases (PDs) are primarily caused by serotypes 19F, 19A, 14, 23F, and 6B [36,37,38]. And our study demonstrated good immunogenicity against these predominant serotypes circulating, with the post-vaccination GMFRs being 5.01, 2.86, 3.94, 5.31 and 6.58 in the total population, respectively.

However, this study also has some limitations. First, this study evaluated the short-term immunogenicity and safety of these two vaccines and two vaccination methods; subsequent long-term follow-up study is needed to evaluate the long-term safety and immune persistence. Second, the sample size was limited; therefore, it is possible that not all low-incidence adverse events were comprehensively reported. To accurately monitor low-incidence adverse events, larger-scale safety observation studies are necessary. Taking this into account, we conducted a subsequent study with 2461 participants to assess the safety of concomitant administration of these two vaccines. This safety observation study showed that concomitant administration of the two vaccines was well tolerated, with no grade 3 or more severe adverse reactions nor any vaccine-related serious adverse events observed [39]. Furthermore, a portion of the participants recruited had underlying conditions. Our study did not specifically evaluate the immunogenicity in individuals with underlying conditions, which needs analysis and verification in further research.

## 5. Conclusions

In conclusion, the concomitant administration of IIV4 and PPSV23 demonstrated non-inferior immunogenicity and did not increase safety risks compared to separate administration in adults aged 60 years and older. Therefore, the concomitant administration of the two vaccines can be implemented if needed.

## Figures and Tables

**Figure 1 vaccines-12-00935-f001:**
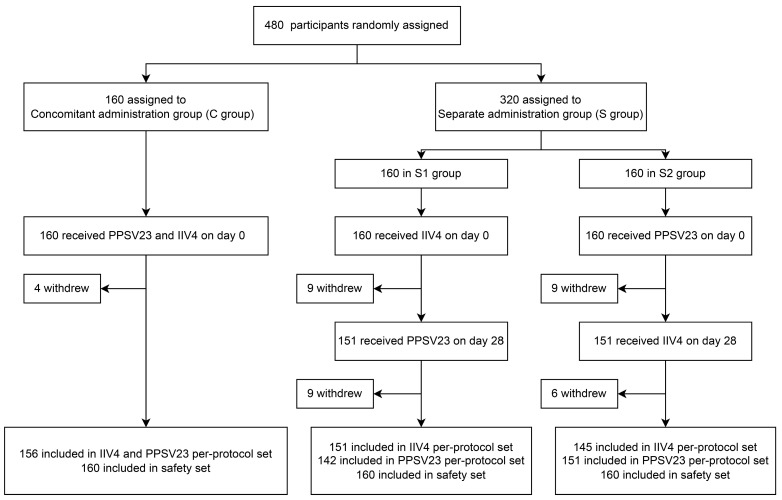
Flowchart of participants’ disposition.

**Figure 2 vaccines-12-00935-f002:**
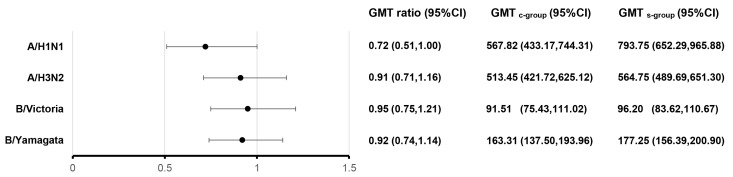
Post-vaccination GMTs and GMT ratios for HI antibodies against four influenza virus subtypes. At 28 days after vaccination with IIV4, administered either concomitantly or alone, the HI antibody GMTs, GMT ratios (GMT_c-group_/GMT_s-group_), and 95%CIs were calculated. GMT: adjusted geometric mean titer; HI: hemagglutination inhibition; IIV4: inactivated quadrivalent influenza vaccine; CI: confidence interval.

**Figure 3 vaccines-12-00935-f003:**
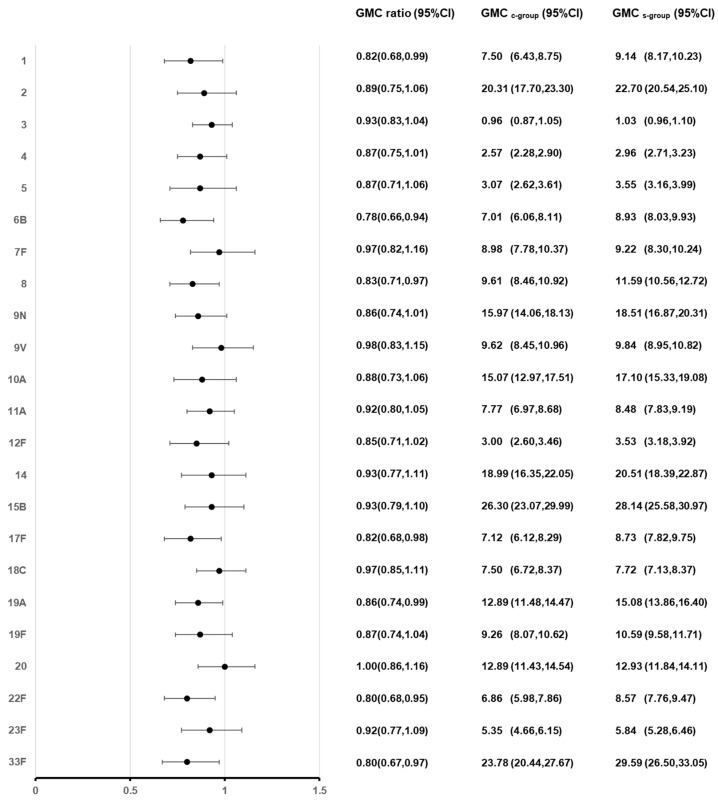
Post-vaccination GMCs and GMC ratios for pneumococcal antibodies of 23 serotypes. At 28 days after vaccination with PPSV23, administered either concomitantly or alone, antibody GMCs, GMC ratios (GMC_c-group_/GMC_s-group_) and 95% CIs were calculated. GMC: adjusted geometric mean concentration; CI: confidence interval.

**Table 1 vaccines-12-00935-t001:** Demographic characteristics and health conditions of participants.

Characteristics		C Group (N = 160)	S Group (N = 320)	Total (N = 480)	*p*-Value
Age, years *		68.36 ± 5.30	68.71 ± 5.35	68.59 ± 5.33	0.5058
Age groups, years	60–69 *n* (%)	106 (66.25%)	192 (60.00)	298 (62.08)	0.1871
	70–79 *n* (%)	49 (30.63%)	116 (36.25)	165 (34.38)	
	80–89 *n* (%)	5 (3.13%)	12 (3.75)	17 (3.54)	
Gender	Male *n* (%)	74 (46.25)	154 (48.13)	228 (47.50)	0.6982
	Female *n* (%)	86 (53.75)	166 (51.88)	252 (52.50)	
Ethnicity	Han *n* (%)	160 (100.00)	320 (100.00)	480 (100.00)	>0.999
Height (cm) *		162.15 ± 8.37	161.43 ± 8.58	161.67 ± 8.51	0.3846
Weight (kg) *		65.91 ± 11.41	65.28 ± 10.50	65.49 ± 10.80	0.5474
BMI *		24.98 ± 3.27	25.01 ± 3.29	25.00 ± 3.28	0.9247

* Mean ± standard deviation. C group: concomitant administration group; S group: separate administration group.

**Table 2 vaccines-12-00935-t002:** Pre-vaccination antibodies level of four influenza virus strains.

Subtype		C Group		S Group		Total		*p* Value
		(N = 156)		(N = 296)		(N = 452)		
		Value	95% CI	Value	95% CI	Value	95% CI	
A/H1N1	SPR, % (*n*)	22.44 (35)	16.15–29.80	20.61 (61)	16.15–25.67	21.24 (96)	17.56–25.30	0.6515
	GMT	12.71	10.70–15.10	12.15	10.69–13.79	12.34	11.14–13.66	0.6769
A/H3N2	SPR, % (*n*)	82.05 (128)	75.11–87.73	84.12 (249)	79.45–88.09	83.41 (377)	79.65–86.72	0.5738
	GMT	59.93	52.20–68.81	71.66	64.70–79.37	67.37	62.06–73.15	0.0420
B/Victoria	SPR, % (*n*)	12.82 (20)	8.01–19.10	17.23 (51)	13.11–22.02	15.71 (71)	12.48–19.40	0.2207
	GMT	13.53	12.01–15.24	13.06	11.85–14.39	13.22	12.26–14.25	0.6634
B/Yamagata	SPR, % (*n*)	48.72 (76)	40.65–56.84	54.39 (161)	48.53–60.17	52.43 (237)	47.72–57.12	0.2508
	GMT	28.92	25.74–32.49	34.27	31.24–37.59	32.32	30.05–34.76	0.0292

C group: concomitant administration group; S group: separate administration group; SPR: seroprotection rate (≥1:40); GMT: geometric mean titer.

**Table 3 vaccines-12-00935-t003:** Post-vaccination antibody levels of four influenza virus strains.

Subtype		C Group		S Group		Total		*p* Value
		(N = 156)		(N = 296)		(N = 452)		
		Value	95%CI	Value	95%CI	Value	95%CI	
A/H1N1	SPR, % (*n*)	91.67 (143)	86.17–95.49	92.57 (274)	88.96–95.28	92.26 (417)	89.40–94.55	0.7333
	SCR, % (*n*)	89.74 (140)	83.88–94.02	91.55 (271)	87.78–94.46	90.93 (411)	87.90–93.41	0.5240
	GMFR	45.66	34.78–59.94	64.60	52.87–78.94	57.31	48.76–67.35	0.0445
A/H3N2	SPR, % (*n*)	100.00 (156)	97.66–100.00	97.97 (290)	95.64–99.25	98.67 (446)	97.13–99.51	0.0977
	SCR, % (*n*)	76.92 (120)	69.51–83.28	75.68 (224)	70.38–80.45	76.11 (344)	71.90–79.97	0.7675
	GMFR	8.33	6.74–10.28	8.00	6.77–9.45	8.11	7.12–9.24	0.7751
B/Victoria	SPR, % (*n*)	83.33 (130)	76.54–88.81	82.77 (245)	77.98–86.89	82.96 (375)	79.18–86.32	0.8797
	SCR, % (*n*)	72.44 (113)	64.72–79.28	73.31 (217)	67.89–78.26	73.01 (330)	68.66–77.05	0.8421
	GMFR	6.85	5.65–8.29	7.32	6.29–8.51	7.15	6.35–8.05	0.6008
B/Yamagata	SPR, % (*n*)	94.23 (147)	89.33–97.33	95.95 (284)	93.03–97.89	95.35 (431)	92.99–97.10	0.4101
	SCR, % (*n*)	69.87 (109)	62.02–76.95	71.28 (211)	65.76–76.37	70.8 (320)	66.37–74.95	0.7536
	GMFR	5.41	4.58–6.40	5.29	4.57–6.11	5.33	4.77–5.95	0.8344

C group: concomitant administration group; S group: separate administration group; SPR: seroprotection rate (≥1:40); SCR: seroconversion rate; GMFR: geometric mean fold rise.

**Table 4 vaccines-12-00935-t004:** Pre-vaccination antibody levels of 23 pneumococcal serotypes.

Serotype		C Group		S Group		Total		*p* Value
		(N = 156)		(N = 293)		(N = 449)		
		Value	95%CI	Value	95%CI	Value	95%CI	
1	GMC (µg/mL)	1.12	0.97–1.28	1.16	1.05–1.29	1.15	1.06–1.24	0.6542
2	GMC (µg/mL)	3.05	2.57–3.61	3.01	2.68–3.39	3.03	2.75–3.33	0.9192
3	GMC (µg/mL)	0.45	0.38–0.52	0.44	0.40–0.49	0.44	0.41–0.48	0.8945
4	GMC (µg/mL)	0.65	0.57–0.73	0.65	0.60–0.71	0.65	0.61–0.70	0.9577
5	GMC (µg/mL)	0.45	0.39–0.51	0.48	0.44–0.52	0.47	0.43–0.50	0.3946
6B	GMC (µg/mL)	1.19	0.99–1.42	1.28	1.12–1.46	1.25	1.12–1.39	0.4999
7F	GMC (µg/mL)	1.14	0.99–1.30	1.32	1.19–1.47	1.26	1.16–1.36	0.0828
8	GMC (µg/mL)	1.78	1.57–2.03	1.92	1.75–2.11	1.87	1.73–2.02	0.3687
9N	GMC (µg/mL)	2.61	2.32–2.94	2.72	2.47–3.00	2.68	2.49–2.90	0.6173
9V	GMC (µg/mL)	1.81	1.57–2.09	1.98	1.79–2.19	1.92	1.77–2.08	0.3141
10A	GMC (µg/mL)	1.96	1.71–2.25	2.18	1.97–2.42	2.10	1.94–2.28	0.2232
11A	GMC (µg/mL)	2.46	2.14–2.83	2.46	2.24–2.70	2.46	2.28–2.66	0.9933
12F	GMC (µg/mL)	1.08	0.94–1.23	0.76	0.69–0.85	0.86	0.79–0.94	0.0002
14	GMC (µg/mL)	4.79	4.22–5.44	5.22	4.66–5.85	5.07	4.65–5.52	0.3167
15B	GMC (µg/mL)	3.93	3.43–4.51	4.72	4.26–5.23	4.43	4.08–4.81	0.0374
17F	GMC (µg/mL)	1.18	1.00–1.39	1.31	1.16–1.48	1.27	1.15–1.39	0.2920
18C	GMC (µg/mL)	2.04	1.82–2.30	2.11	1.94–2.30	2.09	1.95–2.23	0.6465
19A	GMC (µg/mL)	4.92	4.40–5.51	5.04	4.68–5.41	5.00	4.70–5.31	0.7321
19F	GMC (µg/mL)	2.07	1.82–2.35	1.99	1.81–2.19	2.02	1.87–2.18	0.6456
20	GMC (µg/mL)	3.93	3.54–4.36	4.16	3.82–4.52	4.08	3.82–4.35	0.4236
22F	GMC (µg/mL)	1.60	1.43–1.80	1.66	1.51–1.83	1.64	1.53–1.77	0.6522
23F	GMC (µg/mL)	0.97	0.83–1.14	1.12	1.00–1.26	1.07	0.97–1.17	0.1473
33F	GMC (µg/mL)	3.17	2.75–3.65	2.89	2.62–3.20	2.99	2.75–3.24	0.3044

C group: concomitant administration group; S group: separate administration group; GMC: geometric mean concentration.

**Table 5 vaccines-12-00935-t005:** Post-vaccination antibody levels of 23 pneumococcal serotypes.

Serotype		C Group		S Group		Total		*p* Value
		(N = 156)		(N = 293)		(N = 449)		
		Value	95%CI	Value	95%CI	Value	95%CI	
1	GMFR	6.59	5.64–7.69	7.96	7.10–8.93	7.45	6.80–8.18	0.0554
	2-fold increase rate, % (*n*)	90.38 (141)	84.64–94.52	91.81 (269)	88.06–94.68	91.31 (410)	88.32–93.75	0.6099
2	GMFR	6.70	5.70–7.86	7.51	6.77–8.34	7.22	6.61–7.88	0.2210
	2-fold increase rate, % (*n*)	91.67 (143)	86.17–95.49	93.17 (273)	89.65–95.78	92.65 (416)	89.83–94.89	0.5600
3	GMFR	2.15	1.93–2.41	2.33	2.16–2.50	2.27	2.13–2.41	0.2326
	2-fold increase rate, % (*n*)	43.59 (68)	35.68–51.75	51.88 (152)	45.99–57.72	49 (220)	44.28–53.73	0.0944
4	GMFR	3.96	3.47–4.51	4.55	4.17–4.97	4.34	4.03–4.66	0.0748
	2-fold increase rate, % (*n*)	79.49 (124)	72.29–85.53	86.01 (252)	81.50–89.77	83.74 (376)	80.00–87.03	0.0746
5	GMFR	6.61	5.57–7.85	7.58	6.77–8.49	7.23	6.57–7.95	0.1782
	2-fold increase rate, % (*n*)	87.82 (137)	81.64–92.51	91.47 (268)	87.66–94.40	90.2 (405)	87.07–92.79	0.2158
6B	GMFR	5.69	4.88–6.64	7.12	6.37–7.95	6.58	6.02–7.20	0.0200
	2-fold increase rate, % (*n*)	87.82 (137)	81.64–92.51	90.44 (265)	86.48–93.56	89.53 (402)	86.32–92.21	0.3873
7F	GMFR	7.32	6.34–8.46	7.25	6.50–8.08	7.27	6.67–7.93	0.9125
	2-fold increase rate, % (*n*)	93.59 (146)	88.53–96.88	92.15 (270)	88.45–94.96	92.65 (416)	89.83–94.89	0.5778
8	GMFR	5.24	4.54–6.04	6.12	5.54–6.77	5.80	5.34–6.29	0.0752
	2-fold increase rate, % (*n*)	86.54 (135)	80.16–91.47	91.47 (268)	87.66–94.40	89.76 (403)	86.57–92.40	0.1010
9N	GMFR	6.03	5.23–6.95	6.85	6.17–7.60	6.55	6.02–7.13	0.1576
	2-fold increase rate, % (*n*)	87.82 (137)	81.64–92.51	89.42 (262)	85.32–92.70	88.86 (399)	85.58–91.62	0.6080
9V	GMFR	5.12	4.42–5.92	5.07	4.59–5.60	5.09	4.69–5.52	0.9186
	2-fold increase rate, % (*n*)	85.9 (134)	79.43–90.95	85.67 (251)	81.12–89.47	85.75 (385)	82.17–88.85	0.9466
10A	GMFR	7.23	6.25–8.37	8.09	7.23–9.05	7.78	7.12–8.50	0.2371
	2-fold increase rate, % (*n*)	91.67 (143)	86.17–95.49	91.47 (268)	87.66–94.40	91.54 (411)	88.57–93.94	0.9425
11A	GMFR	3.16	2.81–3.56	3.45	3.15–3.77	3.34	3.12–3.59	0.2507
	2-fold increase rate, % (*n*)	69.23 (108)	61.35–76.36	72.7 (213)	67.21–77.72	71.49 (321)	67.07–75.63	0.4386
12F	GMFR	3.18	2.82–3.59	4.31	3.80–4.88	3.88	3.54–4.25	0.0006
	2-fold increase rate, % (*n*)	70.51 (110)	62.69–77.53	78.5 (230)	73.35–83.06	75.72 (340)	71.48–79.62	0.0602
14	GMFR	3.83	3.27–4.50	4.00	3.55–4.50	3.94	3.58–4.33	0.6820
	2-fold increase rate, % (*n*)	66.67 (104)	58.68–74.00	70.99 (208)	65.43–76.12	69.49 (312)	65.00–73.72	0.3435
15B	GMFR	6.27	5.41–7.25	6.17	5.56–6.86	6.21	5.70–6.76	0.8731
	2-fold increase rate, % (*n*)	87.82 (137)	81.64–92.51	88.05 (258)	83.78–91.54	87.97 (395)	84.60–90.83	0.9421
17F	GMFR	5.78	4.90–6.80	6.81	6.05–7.66	6.43	5.85–7.08	0.1065
	2-fold increase rate, % (*n*)	84.62 (132)	77.98–89.89	86.69 (254)	82.26–90.36	85.97 (386)	82.41–89.05	0.5468
18C	GMFR	3.62	3.19–4.11	3.68	3.39–4.00	3.66	3.42–3.92	0.8191
	2-fold increase rate, % (*n*)	76.28 (119)	68.82–82.72	81.91 (240)	77.02–86.15	79.96 (359)	75.95–83.56	0.1560
19A	GMFR	2.59	2.29–2.92	3.01	2.77–3.28	2.86	2.67–3.06	0.0395
	2-fold increase rate, % (*n*)	54.49 (85)	46.33–62.47	68.94 (202)	63.30–74.20	63.92 (287)	59.29–68.37	0.0024
19F	GMFR	4.56	3.92–5.31	5.27	4.77–5.81	5.01	4.61–5.44	0.1070
	2-fold increase rate, % (*n*)	81.41 (127)	74.41–87.18	85.67 (251)	81.12–89.47	84.19 (378)	80.48–87.44	0.2393
20	GMFR	3.17	2.81–3.58	3.17	2.90–3.46	3.17	2.95–3.40	0.9986
	2-fold increase rate, % (*n*)	66.67 (104)	58.68–74.00	68.26 (200)	62.59–73.55	67.71 (304)	63.16–72.01	0.7311
22F	GMFR	4.20	3.65–4.84	5.20	4.69–5.77	4.83	4.44–5.25	0.0161
	2-fold increase rate, % (*n*)	76.28 (119)	68.82–82.72	88.05 (258)	83.78–91.54	83.96 (377)	80.24–87.24	0.0012
23F	GMFR	5.18	4.41–6.07	5.39	4.86–5.98	5.31	4.87–5.80	0.6633
	2-fold increase rate, % (*n*)	85.9 (134)	79.43–90.95	85.67 (251)	81.12–89.47	85.75 (385)	82.17–88.85	0.9466
33F	GMFR	7.83	6.69–9.17	10.00	8.92–11.21	9.19	8.37–10.08	0.0133
	2-fold increase rate, % (*n*)	91.67 (143)	86.17–95.49	94.2 (276)	90.87–96.58	93.32 (419)	90.60–95.45	0.3064

C group: concomitant administration group; S group: separate administration group; GMFR: geometric mean fold rise.

**Table 6 vaccines-12-00935-t006:** The incidences and severities of adverse events.

Adverse Events	C Group (N = 160)	S Group (N = 320)	Total (N = 480)	*p*-Value
Overall	1 (0.63)	5 (1.56)	6 (1.25)	0.6687
Grade 1	1 (0.63)	5 (1.56)	6 (1.25)	0.6687
Grade 2	0 (0.00)	1 (0.31)	1 (0.21)	>0.9999
Local	1 (0.63)	4 (1.25)	5 (1.04)	0.6691
Grade 1	1 (0.63)	4 (1.25)	5 (1.04)	0.6691
Pain	1 (0.63)	4 (1.25)	5 (1.04)	0.6691
Grade 1	1 (0.63)	4 (1.25)	5 (1.04)	0.6691
Induration/Swelling	0 (0.00)	1 (0.31)	1 (0.21)	>0.9999
Grade 1	0 (0.00)	1 (0.31)	1 (0.21)	>0.9999
Redness	0 (0.00)	2 (0.63)	2 (0.42)	0.5546
Grade 1	0 (0.00)	2 (0.63)	2 (0.42)	0.5546
Systemic	0 (0.00)	1 (0.31)	1 (0.21)	>0.9999
Grade 1	0 (0.00)	1 (0.31)	1 (0.21)	>0.9999
Grade 2	0 (0.00)	1 (0.31)	1 (0.21)	>0.9999
Cough	0 (0.00)	1 (0.31)	1 (0.21)	>0.9999
Grade 1	0 (0.00)	1 (0.31)	1 (0.21)	>0.9999
Fever	0 (0.00)	1 (0.31)	1 (0.21)	>0.9999
Grade 2	0 (0.00)	1 (0.31)	1 (0.21)	>0.9999
Fatigue	0 (0.00)	1 (0.31)	1 (0.21)	>0.9999
Grade 2	0 (0.00)	1 (0.31)	1 (0.21)	>0.9999

All data are *n* (%). C group: concomitant administration group; S group: separate administration group.

## Data Availability

The datasets presented in this article are not readily available because the data are part of an ongoing study. Requests to access the datasets should be directed to the corresponding author.

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
