# Peer review of "Immunogenicity and Safety of an Inactivated Quadrivalent Influenza Vaccine Administered Concomitantly with a 23-Valent Pneumococcal Polysaccharide Vaccine in Adults Aged 60 Years and Older"

_vaccines, 2024, doi:10.3390/vaccines12080935_

Round 1

Reviewer 1 Report

Comments and Suggestions for Authors

Zhu et al. investigated the immunogenicity and safety of the concomitant administration of IIV4 and PPSV23 vaccines in the Chinese population. Participants were grouped to C and S groups and their antibodies titer were evaluated in HI assay. The authors demonstrated no serious adverse events in the participants, and they concluded that the immunogenicity of the concomitant administration of IIV4 and PPSV23 was non-inferior to that of the separate administration. This study provides new insights to the concomitant administration of vaccines in elders.   Please see the suggestions below.

1.the authors evaluated the HI titers against influenza strains, so HI assay was performed in this study. please give a description about the assay in material and method section.

2. table 2, figure 1 and table 3 used A/H1N1, A/H3N2 as strains, which is inappropriate. H1N1, H3N2  and B/Yamagata are subtypes, not strains. Please show the stain name of the virus tested.

3. table 2 and 3, it seems that only one strain from each subtype was selected for testing, it’s safer to reach a conclusion if more strains in this subtype can be tested.

Author Response

Comments 1: the authors evaluated the HI titers against influenza strains, so HI assay was performed in this study. please give a description about the assay in material and method section.

Response 1:Thank you very much for pointing this out. We agree with this comment. We have added the descriptions of serological assays in material and method section as subsection 2.6 in line 182-193, page 4, which have been marked in red.

"The antibody titers against four influenza vaccine-associated strains (A/Victoria/2570/2019 (H1N1) pdm09-like virus; A/Cambodia/e0826360/2020 (H3N2)-like virus; B/Washington/02/2019 (B/Victoria lineage)-like virus; B/Phuket/2027/2013 (B/Yamagata lineage)-like virus)) were measured using a hemagglutination inhibition (HI) assay following the WHO Manual for the Laboratory Diagnosis and Virological Surveillance of Influenza [26]. The international reference standards of influenza antigens were obtained from the National Institute for Biological Standards and Control (NIBSC). Pneumococcal antibodies were quantified by enzyme linked immunosorbent assayv(ELISA) according to the WHO manual for the quantitation of Streptococcus pneumoniae serotype specific IgG (Pn PS ELISA) [27]. The pneumococcal polysaccharide standards were sourced from American Type Culture Collection (ATCC)."

Comments 2: table 2, figure 1 and table 3 used A/H1N1, A/H3N2 as strains, which is inappropriate. H1N1, H3N2 and B/Yamagata are subtypes, not strains. Please show the stain name of the virus tested.

Response 2:Thank you very much for pointing this out. We agree with this comment. We have added the strains name in subsection 2.6 in line 183-196, page 4, which have been marked in red. And we have corrected the “strains” into “subtypes” in table 2, figure 2 and table 3.

"The antibody titers against four influenza vaccine-associated strains (A/Victoria/2570/2019 (H1N1) pdm09-like virus; A/Cambodia/e0826360/2020 (H3N2)-like virus; B/Washington/02/2019 (B/Victoria lineage)-like virus; B/Phuket/2027/2013 (B/Yamagata lineage)-like virus)) were measured using a hemagglutination inhibition (HI) assay"

Comments 3: table 2 and 3, it seems that only one strain from each subtype was selected for testing, it’s safer to reach a conclusion if more strains in this subtype can be tested.

Response 3: Thank you for your comments.Your suggestion is excellent. In our immunogenicity study, we selected the WHO-recommended influenza vaccine strains for each subtype to evaluate the basic immunogenicity of the vaccine. We did not assess the immune response to other non-vaccine-related strains. We fully agree that your suggestion is crucial for enhancing the study's comprehensiveness and would indeed make our conclusions more robust. However, the ethics committee for this clinical trial stipulated that blood samples be used solely for testing serum antibodies against the four vaccine-related strains. The samples cannot be used for other tests or research.  Therefore, in this study, we are unable to test or evaluate the immune response of the influenza vaccine against other non-vaccine-related strains.

We strongly agree with you and have incorporated this suggestion into our future research programme, and we plan to prioritise this experiment in our future work to further validate and deepen our findings.

Reviewer 2 Report

Comments and Suggestions for Authors

The peer-reviewed work examined the immunogenicity and reactogenicity of quadrivalent influenza vaccine (IIV4) and the 23-valent pneumococcal polysaccharide vaccine (PPSV23) when administered either concomitantly or separately. Data obtained from the blinded, randomized, placebo-controlled trial included 480 participants older 60 years those were randomly assigned to receive IIV4 and PPSV23 concomitantly (C group), IIV4 on day 0 and PPSV23 on day  28 (S1 subgroup) or PPSV23 on day 0 and IIV4 on day  28 (S2 subgroup). The immunogenicity was assessed using hemagglutination inhibition (HAI) titers. For safety assessment, participants were required to report any adverse events within 28 days following each vaccination. It has been shown, that the concomitant administration of IIV4 and PPSV23 had non-inferior immunogenicity and did not increase safety risks compared to separate administration.

Note:

1)     The study included three groups of participants, however, the analysis and conclusions had compared the first group (C group) with the total data of the second and third groups (S1 and S2 subgroup). Meanwhile, a comparison of all three groups would be more informative, as it would allow comparison of the immune response to each vaccine when administered alone, when administered in combination with another vaccine, and when administered shortly after vaccination with another vaccine

2)       There is an error in the lower right cell of figure 1. Should be: “151 included in IIV4 per-protocol set”

Author Response

Comments 1: The study included three groups of participants, however, the analysis and conclusions had compared the first group (C group) with the total data of the second and third groups (S1 and S2 subgroup). Meanwhile, a comparison of all three groups would be more informative, as it would allow comparison of the immune response to each vaccine when administered alone, when administered in combination with another vaccine, and when administered shortly after vaccination with another vaccine.

Response 1:Thank you very much for your careful review and valuable comments on our paper. We are glad that your point about comparing the data from the three groups is in line with our thoughts. Although the primary outcome of this study was to assess the non-inferiority of immunogenicity in the C group versus the s group, we also attempted to perform post hoc statistical analyses of the three groups of data. However,  our clinical sample size was estimated by the primary objective, the data analyses of the three groups may not be able to reach a valid level of confidence, and therefore we did not tabulate the results of the three-group comparisons in the main text, but rather in the supplementary materials. In order to better represent the similarities and differences in antibody levels between the concomitant group and the two single-injection groups, we added additional detailed descriptions in the main text  in lines 234-240, page 6, and in lines 283-295, page 8,  which have been marked in red. And we presented all the statistical results of the three-group comparisons in tables S3 and S6 of the supplementary materials.

Comments 2:There is an error in the lower right cell of figure 1. Should be: “151 included in IIV4 per-protocol set”.

Response:We sincerely thank the reviewer for careful reading. We have written the wrong number of participants in the PPSV23 per-protocol set. We have corrected the “145 included in PPSV23 per-protocol set” into “151 included in PPSV23 per-protocol set” in figure 1.

Reviewer 3 Report

Comments and Suggestions for Authors

In this manuscript, Zhu et al. determine the safety and immunogenicity of giving the influenza vaccine at the same time as the Pneumococcal vaccine in an aged (60+) population versus giving the 2 vaccines at separate times. This is an important study, as barriers to receiving healthcare are all over the world, and being able to get 2 vaccines during the same appointment can help overcome those barriers, especially in a older population that may have mobility/transportation barriers, and often does not respond as well to vaccines. The authors have done thorough work and do not make any exaggerated claims. This reviewer thinks that the manuscript can be published as is, but will read/consider any comments from other reviewers.

Author Response

Dear Reviewer,

I hope this message finds you well.

We wanted to extend our sincere gratitude for taking the time to review our manuscript, titled "[Immunogenicity and Safety of an Inactivated Quadrivalent Influenza Vaccine Administered Concomitantly with a 23-Valent Pneumococcal Polysaccharide Vaccine in Adults Aged 60 Years and Older]." We are delighted to learn that there were no revisions needed, and we appreciate your positive feedback on my work. 

Thank you once again for your support and for helping me improve my research.

Best regards,

Xi Lu,

Medical Mananger,

Sinovac Biotech Co., Ltd.

Round 2

Reviewer 1 Report

Comments and Suggestions for Authors

This is an updated version, and the authors addressed the concerns